# Single-molecule insights into surface-mediated homochirality in hierarchical peptide assembly

Yumin Chen[1,4], Ke Deng[1], Shengbin Lei[2], Rong Yang[1], Tong Li [3], Yuantong Gu [3], Yanlian Yang [1], Xiaohui Qiu[1] & Chen Wang[1]

Homochirality is very important in the formation of advanced biological structures, but the origin and evolution mechanisms of homochiral biological structures in complex hierarchical process is not clear at the single-molecule level. Here we demonstrate the single-molecule investigation of biological homochirality in the hierarchical peptide assembly, regarding symmetry break, chirality amplification, and chirality transmission. We find that homochirality can be triggered by the chirality unbalance of two adsorption configuration monomers. Co-assembly between these two adsorption configuration monomers is very critical for the formation of homochiral assemblies. The site-specific recognition is responsible for the subsequent homochirality amplification and transmission in their hierarchical assembly. These single-molecule insights open up inspired thoughts for understanding biological homochirality and have general implications for designing and fabricating artificial biomimetic hierarchical chiral materials.

[1] CAS Key Laboratory of Biomedical Effects of Nanomaterials and Nanosafety, CAS Key Laboratory of Standardization and Measurement for Nanotechnology, CAS Center for Excellence in Nanoscience, National Center for Nanoscience and Technology, China, Beijing 100190, China. [2] Department of Chemistry, School of Science & Collaborative Innovation Center of Chemistry Science and Engineering (Tianjin), Tianjin University, Tianjin 300072, China. [3] School of Chemistry, Physics and Mechanical Engineering, Queensland University of Technology, Brisbane 4000 QLD, Australia. [4] Present address: Fujian Institute of Research on the Structure of Matter, Chinese Academy of Sciences, Fuzhou 350002, China. Correspondence and requests for materials should be addressed to K.D. (email: kdeng@nanoctr.cn) or to Y.Y. (email: yangyl@nanoctr.cn) or to X.Q. (email: xhqiu@nanoctr.cn) or to C.W. (email: wangch@nanoctr.cn)

Homochirality is an important selection rule in the formation of living organisms, remaining to be a generally interesting and significant topic for extensive investigations[1]. As widely documented, only L-amino acids are encoded to form proteins and only D-sugars form the backbones of DNA, and these proteins and DNA are mostly right-handed helical structures in biological systems[2–4]. At the macroscopic level, the geometric structures of many living organisms prefer to exhibit homochirality. For example, the majority of gastropod species have right-handed shells[5]. Biological chirality is not only limited to molecular chirality caused by chiral central atom but also includes structural chirality in the topological geometric space such as DNA helical structure. Different from the concept of molecular chirality in organic chemistry; structural chirality emphasizes the rotational structure in topological geometric space rather than focuses on an individual chiral central atom. In biological system, hierarchical assembly is a key strategy to propagate the chirality from lower level (L-amino acids and D-sugars) to its higher levels such as DNA, cells, tissues, and organisms[6–9]. A typical example is the formation of cilia, in which L-polypeptide chains consisted of L-amino acids fold into alpha-helical proteins, further assemble into chiral microtubules, and finally form cilia[10]. Their chirality is delivered from the L-amino acid building blocks to the advanced cilia structure via a four-level assembly. How complex homochiral structures form, and how chiral information propagates from low levels to high levels in the hierarchical assembly process, are definitely significant scientific questions. Up to now, the molecular mechanism about the origin and evolution of homochiral biological structures in the complex hierarchical assembly processes still remains unclear and very challenging.

Using high-resolution detection methods and overcoming the challenge caused by complex biological structures are the key issues to reveal the underlying molecular mechanism. Scanning tunneling microscopy (STM) provides a promising solution to investigate the two-dimensional (2D) surface-mediated structural homochirality at the single-molecule level. On the one hand, well-defined single-crystal substrate effectively reduces the conformation diversity of the adsorbed biological molecules; on the other hand, STM has been proved to be a powerful high-resolution technique in studying the chiral recognition and single-step assembly of amino acid or peptide[11–15]. Recently, great interests have also been drawn in the chirality in the hierarchical assemblies of organic molecules on surfaces[16–18]. However, most of studies indicate that the assembly of two kinds of heterochiral organic molecular building blocks usually produces a mixture of racemic assembled domains instead of only one kind of homochiral assemblies, which is no complying with the homochirality rule in the biological systems. Hence, it is necessary to directly study the chirality in the hierarchical assembly of biomolecules, in order to better understand the biological homochirality.

Herein, we investigate the surface-mediated homochirality evolution process in the hierarchical assembly of valinomycin from single-molecule to supramolecular level. STM combined with density functional theory (DFT) calculations reveal that two kinds of chiral adsorption configuration monomers with unequal amount coexist on the surface. This initial chirality unbalance is amplified by the site-specific recognition between these two heterochiral monomers in the first-level assembly, leading to the formation of homochiral tetramers. The homochirality is further transferred when these homochiral tetramers assembly into the homochiral supramolecular networks. We find that homochirality can be triggered by the chirality unbalance of two adsorption configuration monomers. Co-assembly between these two adsorption configuration monomers is very critical for the

formation of homochiral assemblies. The site-specific recognition is responsible for the subsequent homochirality amplification and transmission in their hierarchical assembly.

## Results

**Surface-mediated chirality unbalance of valinomycin monomers.** Valinomycin was selected as a scientific model owning to its simplified three-fold symmetrical cyclic structural characteristics. As shown in Fig. 1a, valinomycin is a cyclic dodecadepsipeptide, which consists of three repeated asymmetric chiral structural units, L-valine, D-hydroxyvaleric acid, D-valine, and L-lactic acid (L-Val—D-Hyv—D-Val—L-Lac). Valinomycin is a nonplanar macrocyclic molecule having different chemical moieties at both sides of the annular backbone; therefore, it has two different landing faces to contact with surfaces[19,20]. STM observation (Fig. 1b) confirmed that two kinds of adsorption configurations coexist on the Cu(111) surface when valinomycin molecules were deposited on the Cu(111) surface at 78 K. Further DFT calculations reveal that the topological geometric arrangement of three asymmetric tetrapeptide structural units is counterclockwise as the arrows shown, when valinomycin molecule landing on the Cu(111) surface via its A face (Fig. 1c). We called this adsorption configuration valinomycin monomer as L-type configuration monomer ($M_L$) and simplified it as a yellow counterclockwise propeller. On the contrary, if the B face contacts with the Cu(111) surface, the arrangement of three asymmetric tetrapeptide structural units is clockwise as the arrows displayed (Fig. 1d). In this case, it was defined as R-type configuration monomer ($M_R$) and simplified as a blue clockwise propeller. According to the corresponding relationships between molecular models, electron density images and STM images as our previous work presented, $M_L$ is seen as three lobes with a central protrusion, while $M_R$ appears as three lobes with a central cavity in the STM image[20].

DFT calculations reveal that three intramolecular hydrogen bonds form between the carbonyl oxygen of L-Lac residue and the amine group of D-Val residue (Supplementary Figure 1). The energy of a hydrogen bond is 9.34 kcal mol$^{-1}$ and the H•••O distance is 1.81 Å. Hydrogen bonds are strong enough to stabilize the framework of the cyclic main chain in the form of a shallow bowl. The side chains lie on the exterior of the bowl with a certain orientation, which is determined by the chirality of the amino acid residues (L-Val, D-Hyv, D-Val, and L-Lac). Such spatial restriction for valinomycin leads to an unambiguous counterclockwise configuration when valinomycin adsorbed on the surface via the A face. Similarly, the arrangement of three asymmetric tetrapeptide structural units is clockwise, if valinomycin adsorbed on the Cu(111) surface via the B face. Obviously, a gaseous valinomycin has not defined structural chirality when the chiral reference (the Cu(111) surface) is absent, since it flips randomly in three-dimensional (3D) space. Only when valinomycin adsorbed on the surface, it will be immobilized with an unambiguous configuration under the surface confinement. Surface adsorption plays a key role in the formation of 2D structural chirality and the structural chirality is determined by its asymmetrical chiral tetrapeptide units.

It is worth mentioning that the hierarchical assembly does not start from gaseous valinomycin but a pair of enantiomers ($M_L$ and $M_R$). The symmetric mirror of these two adsorption configurations origin from the same molecule is parallel to the Cu(111) surface, which is different from the reported cases in which the symmetric mirror of two initial molecular building block enantiomers is usually perpendicular to the surface[21]. A statistical analysis of a large number of STM images reveals that the quantities of $M_R$ are larger than $M_L$ (Fig. 1e). Though the

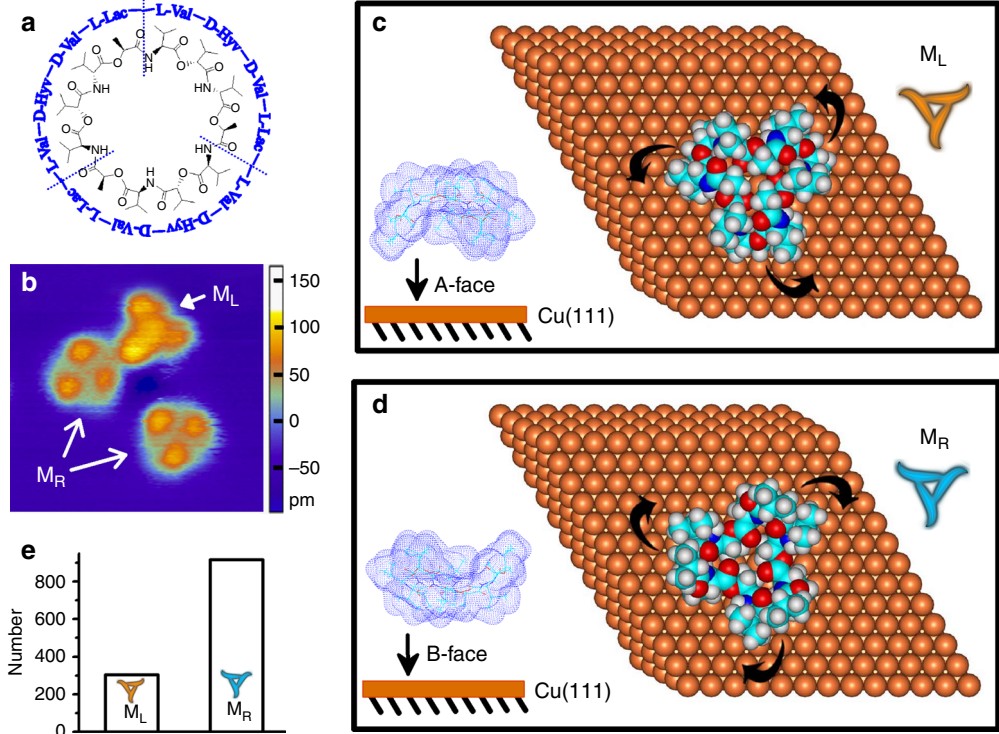

**Fig. 1** Individual valinomycin monomer. **a** Structure formula of valinomycin. **b** STM image (8 nm × 8 nm) of valinomycin monomers adsorbed on Cu(111) surface at 78 K. $M_L$ and $M_R$ coexist on the surface. **c, d** Calculated adsorption configurations of $M_L$ and $M_R$. $M_L$ and $M_R$ were schematically simplified as yellow counterclockwise propeller and blue clockwise propeller in the top right corners, respectively. Side views of simulated molecular model of valinomycin packed with electron density deposited onto the Cu(111) surface were inserted in the bottom left corners. In calculated molecular models, copper, carbon, nitrogen, oxygen, and hydrogen atoms were displayed in brass, cyan, blue, red, and gray, respectively. **e** Statistic molecular numbers of $M_L$ and $M_R$

probabilities to land on the surface via their A face or B face are same, the difference between adsorption and desorption energies will affect the ultimate molecular number of $M_L$ and $M_R$ on the surface. The DFT-calculated adsorption energies indicate that $M_R$ ($-39.646$ kcal mol$^{-1}$) is more stable than $M_L$ ($-33.895$ kcal mol$^{-1}$) (referring to calculation details in the Methods section: a more negative energy indicates that the calculated system is more stable). The calculation results by the DFT-D3 method also indicate that $M_R$ ($-64.203$ kcal mol$^{-1}$) is more stable than $M_L$ ($-57.899$ kcal mol$^{-1}$). As mentioned above, the chemical composition and structure of the two faces of valinomycin are different. As a result, adsorption energy difference accounts for the unequal probabilities of these two adsorption configurations. Desorption of $M_L$ may occur more easily than $M_R$, which results in the higher amount $M_R$ on the surface than $M_L$. The difference in adsorption–desorption thermodynamics results in the quantities unbalance of the two adsorbed configurations with different structural chirality[22,23]. This symmetry break of the chirality is crucial to induce the subsequent homochirality in the hierarchical assembly of valinomycin molecules[2,24,25].

**Chirality amplification at the first-level assembly**. Spontaneous recognition of monomers results in the formation of valinomycin tetramers during the flash annealing of the Cu(111) surface. Typical STM image (Fig. 2a) shows that the adsorbed tetramers have highly symmetrical architectures with three arms on the Cu (111) surface. Comparing the tetramer with the monomers, we found that every tetramer contains a central $M_L$ subunit partially overlapping with three surrounding $M_R$ subunits (Supplementary Figure 2a). The radius of tetramer (3 nm) is less than the sum of

the $M_L$ radius (1.3 nm) and the $M_R$ diameter (2.6 nm), suggesting that the overlap and strong interactions happen between $M_L$ and $M_R$. The bright protrusions at the binding sites of the central $M_L$ subunit with the external $M_R$ subunits of tetramer are about 0.4 Å higher than the other lobes of $M_R$ subunit (Supplementary Figure 2b, c). In contrast, other lobes of the tetramer still look like round protrusions with the same height as the lobes of monomers. The contrast of the apparent heights also indicates that strong site-specific binding and electron overlap takes between $M_L$ and $M_R$. Tetramer can stably exist as a whole after the rotation and translation manipulations on the Cu(111) surface, meaning the interactions between $M_L$ and $M_R$ subunits of tetramer are strong enough (Supplementary Figure 3). DFT calculations were performed to simulate eight typical tetramer models and interactions. The adsorption configurations of a tetramer, subunit-pairing way, and the outer subunit-attacking mode are shown in Fig. 2b–i. The total interaction energies listed in Table 1 reveal that $M_L$–$3M_R$-right is the most stable tetramer model when three outer $M_R$ subunits attack from the right side of the central $M_L$ subunit by head-to-head recognitions. The DFT-D3 method was further performed to estimate the interaction energies, and the results also showed that $M_L$–$3M_R$-right is the most stable tetramer model, which is consistent with the DFT methods (Supplementary Table 1). The electron density is highly consistent with the STM image of tetramer (Fig. 2j), confirming that the tetramer is produced according to the $M_L$–$3M_R$-right model. The electron densities accumulation owing to the strong interactions between $M_L$ and $M_R$ well explains the brighter protrusions at the binding sites in the STM image. A careful examination of the bright protrusions at the binding sites between $M_L$ and $M_R$ subunits, we found that the array of three bright

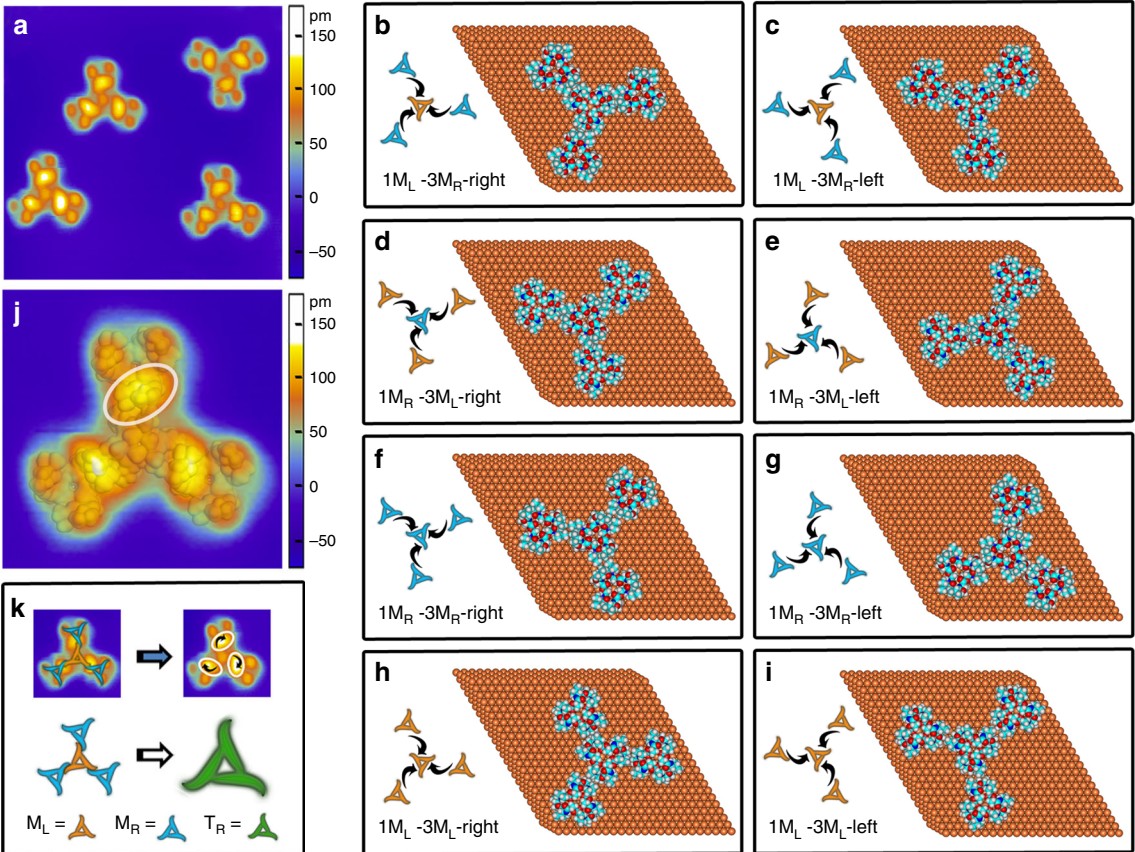

**Fig. 2** Individual valinomycin tetramer. **a** STM image (18 nm × 18 nm) of valinomycin tetramers on Cu(111) surface at 78 K. **b–i** Eight typical calculated molecular models of valinomycin tetramer. The adsorption configurations of tetramer, subunit-pairing way, and outer subunit-attacking mode were shown from **b** to **i**. In calculated molecular models, copper, carbon, nitrogen, oxygen, and hydrogen atoms are displayed in brass, cyan, blue, red, and gray, respectively. **j** STM image (6 nm × 6 nm) of valinomycin tetramers superimposed by calculated electron density. The binding site between $M_L$ and $M_R$ subunits was highlighted by a white circle. **k** Schematic diagram of chirality amplification at the first level of hierarchical assembly

### Table 1 Calculated interaction energies of valinomycin tetramer on the Cu(111) surface by DFT simulations

| Tetramer model | Sum of interaction energy among monomer subunits (kcal mol$^{-1}$) | Tetramer–substrate interaction energy (kcal mol$^{-1}$) | Total interaction energy (kcal mol$^{-1}$) |
|---|---|---|---|
| $M_L$-3$M_R$-right | −26.003 | −145.658 | −171.661 |
| $M_L$-3$M_R$-left | −18.340 | −147.134 | −165.474 |
| $M_R$-3$M_L$-right | −18.287 | −141.604 | −159.891 |
| $M_R$-3$M_L$-left | −20.309 | −136.246 | −156.555 |
| $M_R$-3$M_R$-right | −15.446 | −149.535 | −164.981 |
| $M_R$-3$M_R$-left | −15.424 | −152.180 | −167.604 |
| $M_L$-3$M_L$-right | −14.971 | −125.885 | −140.856 |
| $M_L$-3$M_L$-left | −14.445 | −130.237 | −144.682 |

A more negative energy indicates that the calculated system is more stable

protrusions in the tetramer looks like a rotating three-leaf pinwheel with a chirality. The black arrows marked in Fig. 2k illustrate the clockwise direction, suggesting that the site-specific binding between $M_L$ and $M_R$ is highly directional; as a result, the tetramer is right-handed chiral in geometric topology. The chirality amplification at the first level of the hierarchical assembly was schematically summarized in Fig. 2k. Overlaping the STM image of tetramer with the schematic symbols of the monomers clearly demonstrates that tetramer is produced through the chiral recognition among three outer $M_R$ and one central $M_L$ in a right-attacked head-to-head recognition way. There binding sites between $M_R$ and $M_L$ subunits marked by white circles array like a clockwise rotating three-leaf pinwheel (marked by black arrows),

leading to a right-handed chiral tetramer ($T_R$). The schematic symbols of chiral propeller clearly describe the chirality amplification at the first level assembly. Detailed experimental results further reveal that only one kind of homochiral assemblies ($T_R$) was observed on the Cu(111) surface. In the previously reported examples, a mixture of assemblies' enantiomers was produced since two chiral building blocks self-assemble alone into their corresponding chiral assemblies[16–18,26]. In a word, left-hand building blocks self-assemble into left-hand assemblies, and right-hand building blocks self-assemble into right-hand assemblies. However, in this report, valinomycin tetramers, either $T_L$ or $T_R$, are the co-assembled products of two kinds of adsorbed configuration monomers ($M_L$ and $M_R$). Different from the undisturbed

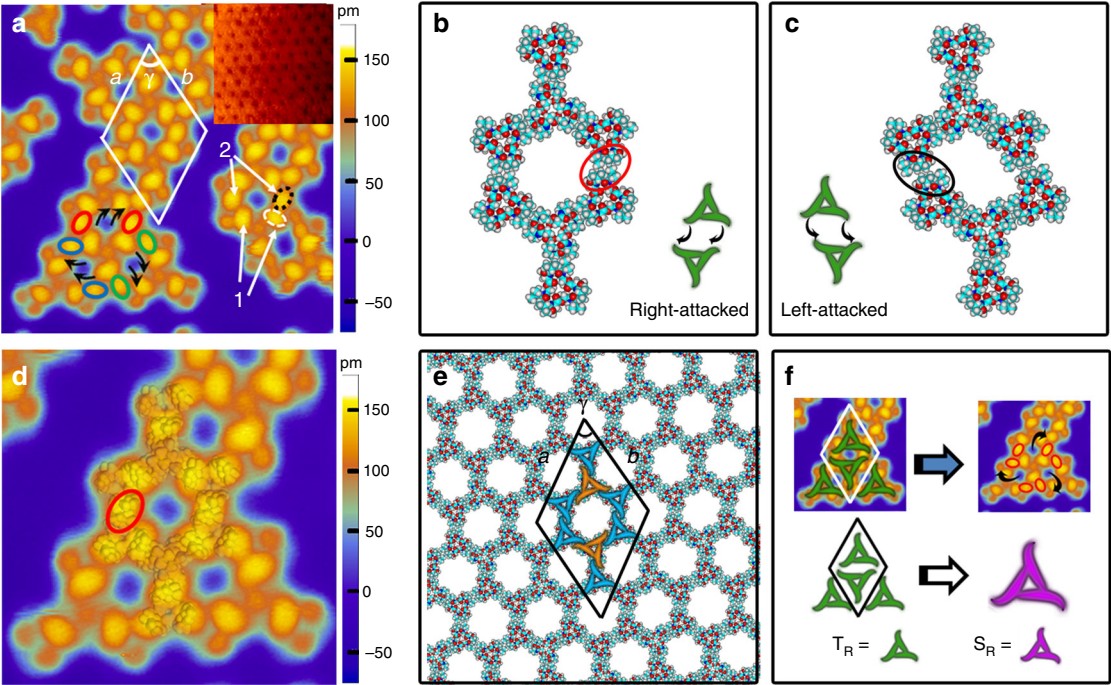

**Fig. 3** Valinomycin supramolecular networks. **a** High-resolution STM image (20 nm × 20 nm) of valinomycin supramolecular networks. As supplement, STM image (25 nm × 25 nm) of an intact valinomycin supramolecular network was also inserted in the top right corner. The directional binding of tetramer building blocks are marked by three groups of oval circles with red, blue, and green colors, and black arrows. The binding sites at the first and second level of assemblies were marked by white and black dashed circles, respectively. **b, c** Two typical calculated molecular models of $T_R$ pair. Right-attacked mode and left-attacked modes were displayed in **b** and **c**, respectively. Their binding sites between $T_R$ subunits were separately highlighted by a red circle and a black circle. **d** STM image (12 nm × 12 nm) of valinomycin supramolecular network superimposed by the calculated electron density of $T_R$ pair. The binding site between $T_R$ subunits was highlighted by a red circle. **e** Calculated molecular model of valinomycin supramolecular networks. A schematic unit cell was superimposed on the model of the supramolecular networks. **f** Schematic diagram of the chirality transmission at the second level of hierarchical assembly. In calculated molecular models, carbon, nitrogen, oxygen, and hydrogen atoms are displayed in cyan, blue, red and gray, respectively. Note: In order to clearly display the calculated molecular models of valinomycin networks, the Cu(111) surfaces have not been shown

self-assembly, the formation of $T_L$ and $T_R$ are competitive. Our calculated results in Table 1 indicate that the interaction energy among monomer subunits in the formation of $T_R$ is more stable than that of $T_L$. The directional site binding in the right-attacked chiral recognition becomes the predominant chiral recognition mode, and $T_R$ becomes the dominant assembly product. We believe that the spatial conformational complementary and the strong intermolecular interactions play an important role in the formation of homochiral tetramers. In a word, the chiral recognition between two converse chiral configuration monomers amplifies the initial chirality unbalance and leads to the homochiral valinomycin tetramers.

**Homochirality transfer at the second-level assembly**. The tetramers formed at the first level act as the building blocks to form supramolecular networks at the second level of hierarchical assembly at a higher coverage of valinomycin molecules. An intact valinomycin supramolecular network was displayed in the inset of Fig. 3a, and more details about the supramolecular networks were revealed in the high-resolution STM image. The unit cell of the supramolecular networks includes two side-by-side inverted $T_R$ with parameters $a = b = 6.0 \pm 0.1$ nm and $\gamma = 60°$. Different from the single site-specific binding between heterochiral pairs ($M_L$ and $M_R$) in the formation of $T_R$, double site-specific binding takes place between homochiral pairs ($T_R$) in the formation of supramolecular networks. We found that three groups of binding-site pairs (marked as red, blue, and green oval circles in Fig. 3a) around a tetramer are tilted in a clockwise

rotation array as the black arrows show, resulting in supramolecular networks with right-handed chirality ($S_R$) in geometric topology. In a word, the directional site-specific binding among homochiral $T_R$ leads to the formation of $S_R$. As a result, homochirality is transferred from $T_R$ to $S_R$ at the second level of hierarchical assembly. DFT calculations were performed to simulate the side-by-side recognitions between $T_R$. Typical right-attacked and left-attacked modes are displayed in Fig. 3b, c. The interaction energies of right-attacked and left-attacked $T_R$ pairs are $-11.778$ and $-5.122$ kcal mol$^{-1}$, respectively, indicating that $T_R$ prefers to self-assemble into $S_R$ through the right-attacked mode. The binding sites between $T_R$ (marked by red circle in Fig. 3b) in the right-attacked mode are obviously matched with that in the STM image (marked by black dashed circle in Fig. 3a), but the binding sites in the left-attacked mode (marked by black circle in Fig. 3c) are different from that in the STM image. The highly agreement between the electron density and STM image further confirmed that right-attacked mode happens and results in the formation of $S_R$. The electron densities accumulation owing to the site-specific binding between $T_R$ well explains the brighter protrusions at the binding sites in the STM image. The calculated lattice parameters for $S_R$ networks (Fig. 3e) are $a = b = 5.97$ nm and $\gamma = 60°$, which agree well with the experimental investigation. The unit cell of hexagonal $S_R$ networks produced from the hierarchical assembly contains two valinomycin tetramers or eight monomers, so that the dimension of the unit cell is far bigger than that in the simple one-step assembly of monomers. Figure 3f schematically demonstrates that $S_R$ is formed through the self-assembly by using homochiral $T_R$ as building blocks. By

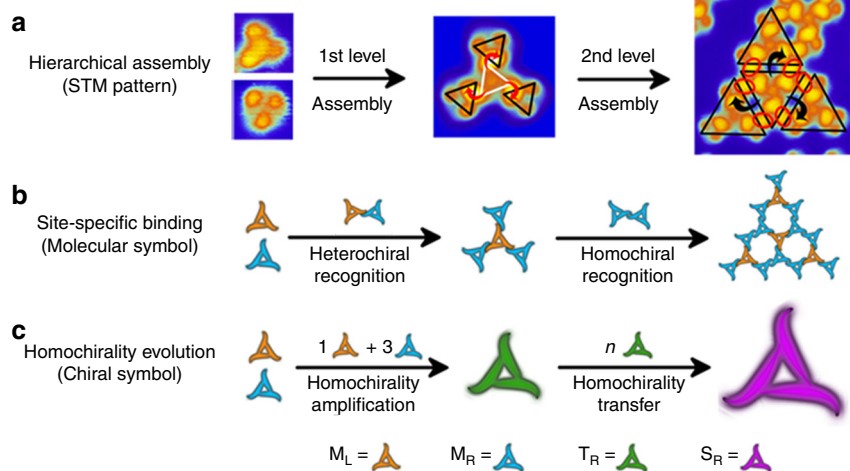

**Fig. 4** Homochirality evolution in the hierarchical assembly. **a** Directional site-specific binding in the hierarchical assembly. The STM image sizes of valinomycin monomer, tetramer, and supramolecular network portion are 3 nm × 3 nm, 6 nm × 6 nm, and 12 nm × 12 nm, respectively. The directional binding sites were highlighted with circles and arrows. **b** Schematic diagram of the recognition way of subunits in the hierarchical assembly. **c** Schematic drawing of the homochirality evolution in the hierarchical assembly

superimposing the symbols of $T_R$ on the STM image of the supramolecular networks, it becomes clear that every $T_R$ subunit is surrounded by three homochiral $T_R$ subunits and the directional right-attached site binding ensures the transmission of the homochirality from $T_R$ to $S_R$ at the second level of the hierarchical assembly.

## Discussions

Figure 4 demonstrates that the formation of hierarchical supramolecular structure is accompanied with the evolution of the homochirality. The STM images in Fig. 4a directly demonstrate that the formation of R-type supramolecular networks ($S_R$) is a perfect hierarchical assembly process. $T_R$ produced by chiral recognition of $M_L$ and $M_R$ at the first level further assemble into $S_R$ at the second level of hierarchical assembly. Site-specific binding between the building blocks is directional at both first and second levels of the hierarchical assembly of valinomycin molecules. Spatial conformational complementary is deemed as an essential condition for the molecular recognition of peptides or proteins[11,27]. We also considered that it is a key factor for homochirality evolution in valinomycin hierarchical assembly. When right-attacked mode is adopted, spatial configuration complementary is advantageous to produce stronger van der Waals interactions in the directional assemblies, which ensures that the homochirality is transferred from one level to another lever in the hierarchical assembly. Though the binding sites at the first and second level both exhibit clockwise arrangement, their geometry and size are different as revealed from the STM images (Fig. 4a). The binding mode and intensity at the different levels are not the same. Comparing the schematic models (Fig. 4b) with the corresponding STM images (Fig. 4a), we found that the single site-specific binding happens between the heterochiral subunits ($M_L$–$M_R$) at the first level, whereas double site-binding takes place between the homochiral subunits ($M_R$–$M_R$) at the second level. Our DFT calculations reveal that the heterochiral pair interaction is stronger than the homochiral pairs, indicating the interaction at the first level is stronger than that at the second level in the hierarchical assembly of valinomycin[28]. Figure 4c demonstrates the evolution process of the homochirality in the hierarchical assembly of valinomycin. Asymmetrical adsorption of the nonplanar valinomycin molecules on the Cu(111) surface leads to unequal population of $M_L$ and $M_R$, which is the initiate

building blocks for the hierarchical assembly. The initial symmetry break is amplified by the directional site-specific recognition between these two heterochiral monomers at the first-level assembly, leading to the formation of homochiral tetramers ($T_R$). The homochirality is further transferred when these homochiral tetramers assembly into the homochiral supramolecular networks ($S_R$) in the second level assembly.

This work demonstrated how a biomolecule produces two adsorption configurations ($M_L$ and $M_R$) with a diverse 2D structural chirality and then assembles into a homochiral tetramer and supramolecular networks ($T_R$ and $S_R$) on the achiral Cu (111) surface. The hierarchical assembly starts by using $M_L$ and $M_R$ as the initial building blocks. The homochirality in the hierarchical assembly is originated from the chirality unbalance of two adsorption configurations ($M_L$ and $M_R$), rather than from the gaseous valinomycin. The surface-mediated homochirality can be triggered by the chirality unbalance of two adsorption configurations coming from the same initial biomolecule, instead of using a single homochiral molecule as building blocks, or achiral building blocks guided by proper amount of chiral dopants[24,29]. Our work demonstrates another way to trigger the formation of advanced homochiral biological structures.

This report also provides the single-molecule evidence that hierarchical assembly is a significant strategy for the self-organization of highly ordered biological homochiral structures. The hierarchical assembly strategy can be propagated into the field of the fabrication of versatile 2D artificial chiral molecular structures and materials. The candidate molecules may possess the following features: (1) the molecules have multiple binding sites and symmetries for hierarchical assembly[28]; (2) the molecules can produce two kinds of adsorption configurations with reverse structural chirality when they adsorbed on the surface; and (3) these two kinds of chiral adsorption configurations monomer can co-assembly instead of self-assembly.

In summary, we have demonstrated the origin and evolution of surface-mediated biological homochirality in the hierarchical assembly of valinomycin, regarding symmetry break, chirality amplification, and chirality transmission. Our work provides the single-molecule evidence that hierarchical assembly is a significant strategy for the self-organization of highly ordered biological chiral structures. We found a possible trigger way for the formation of advanced homochiral biological structures and materials, i.e., surface-mediated biological homochirality can

origin from the chirality symmetry break of two adsorption configuration monomers. We also revealed that co-assembly between these two adsorption configuration monomers is very critical for the formation of homochiral assemblies. The directional site-specific recognition is responsible for the homochirality amplification and transmission in the hierarchical assembly. These fresh single-molecule insights open up thoughts for understanding biological homochirality in the complex hierarchical assembly process, and may have helpful inspiration for designing and fabricating artificial biomimetic hierarchical chiral materials.

## Methods

**Experimental details**. The experiments were performed in an Omicron ultrahigh vacuum STM. Cu(111) single crystal was cleaned by repeated cycles of Neon ion bombardment at room temperature and annealing at about 850 K. Valinomycin molecules (Aldrich, 90%) were vapor deposited onto Cu(111) surface at 78 K from a heated crucible, followed by a flashing annealing at room temperature for 2 min. STM experiments were carried out at 78 K, and all images were obtained at a constant current mode with a sample bias of −2 V and a tunneling current of 0.03 nA.

**Calculation details**. Theoretical calculations were performed using DFT provided by the DMol3 code[30]. In DMol3, the electronic wave function was expanded numerically on a dense radial grid. The double-numeric polarized basis sets were used, which is comparable to Gaussian 6-31G* basis sets. In the local spin density approximation, exchange and correlation were described by the Perdew and Wang parameterization of the local exchange-correlation energy[31]. DFT semilocal pseudopotentials were adopted to represent for the inner core electrons of Cu and 19 valence electrons were treated explicitly for Cu ($3s$, $3p$, $3d$, $4s$). Spin-restricted wave functions were employed within a cutoff radius of $R_{cut} = 5.5$ Å. For the self-consistent field procedure, a convergence criterion on the energy and electron density was $10^{-5}$ a.u.

We have adopted a five-layer Cu slab model with the periodic boundary conditions to evaluate the interaction energy between valinomycin and the Cu (111) surface. In the calculation, an optimized Cu lattice with parameter 3.6145 Å was used to reduce the effect of the stress, which is nearly equal to the experimental value (3.6149 Å), suggesting our DFT methods are suitable for this system. In the superlattice, we employed Cu slabs separated by 45 Å in the normal direction. Supercells (20 × 20) were used in modeling the adsorbates on Cu (111) and the Brillouin zone was sampled by a $1 \times 1 \times 1$ $k$-point mesh. The interaction energy $E_{inter}$ is given by $E_{inter} = E_{Cu-valinomycin} - (E_{valinomycin} + E_{Cu})$.

We have further performed the DFT-D method to estimate the interaction energy between adsorbates and Cu(111) surface, in which the London dispersion interaction in van der Waals interaction is included. In surface science, thousands of different systems including intermolecular and intramolecular cases have been investigated by the DFT-D method successfully[32]. Here, we employed the DFT-D3 method based on the standard Kohn–Sham DFT. The corrected energy is added with an atom-pair wise (atom-triple wise) dispersion correction as follows[33]:

$$E_{DFT-D3} = E_{KS-DFT} + E_{disp} \tag{1}$$

In the DFT-D3 method[32], the vdW-energy expression is

$$E_{disp} = -\frac{1}{2} \sum_{i=1}^{Nat} \sum_{j=1}^{Nat} \sum_{L}{}' \left( f_{d,6}(r_{ij,L}) \frac{C_{6ij}}{r_{ij,L}^6} + f_{d,8}(r_{ij,L}) \frac{C_{8ij}}{r_{ij,L}^8} \right) \tag{2}$$

where $r_{ij,L}$ is the internuclear distance between atoms $i$ and $j$. The dispersion coefficients $C_{6ij}$ and $C_{8ij}$ are adjusted and depend on the local geometry around atoms $i$ and $j$. The Becke–Jonson damping is used in the DFT-D3 method:

$$f_{d,n}(r_{ij}) = \frac{S_n r_{ij}^n}{r_{ij}^n + (a_1 R_{0ij} + a_2)^n} \tag{3}$$

where $R_{0ij} = \sqrt{\frac{C_{8ij}}{C_{6ij}}}$, the parameters $a_1$, $a_2$, $s_6$, and $s_8$ are adjustable depending on the choice of exchange-correlation functional.

**Data availability**. The data that support the findings of this study are available from the corresponding authors upon reasonable request.

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

## Acknowledgements

We thank Dr. Huifang Xu (Harbin Institute of Technology, China) and Dr. Xiaobo Mao (Johns Hopkins University, USA) for their helpful discussions. This work was supported by the Ministry of Science and Technology (2017YFA0205000).

## Author contributions

Y.C. designed the project, performed the experiments and wrote the manuscript; C.W. designed the project, discussed the data and modified the manuscript; K.D. performed the theoretical simulation, discussed the data and modified the manuscript; T.L. and Y.G. performed the theoretical simulation and discussed the data; X.Q., Y.Y. and S.L. discussed the data and modified the manuscript; R.Y. discussed the data.

## Additional information

**Competing interests:** The authors declare no competing interests.

