## [Peer Review File · Nature Communications]

Reviewers' comments:

Reviewer #1 (Remarks to the Author):

In the submitted manuscript, Chen et al. reported the formation of homochirality upon adsorption of valinomycin molecules on a Cu(111) surface. The authors claimed to observe a difference in population between the two possible molecular adsorption configurations (A and B faces), leading to the formation of a predominant homochiral species, mimicking the origin of chirality in living organisms.

The initial symmetry breaking is then followed by hierarchical growth of a supramolecular assembly of homochiral structures, revealing how the monomer chirality unbalance is amplified in more complex aggregates.

Despite chirality expression on surfaces has been extensively investigated in the last decades by means of surface science techniques, the reported results seem to bring some new inputs in such research field, in particular for their importance in biological systems.

However, there are a few major discrepancies in the text, that need to be clarified.

Firstly, the formation of chirality through an achiral organic molecule adsorbed on an achiral surface is somehow surprising, probably representing the more interesting results in the whole manuscript. Nevertheless, the authors did not properly comment on the origin of such behavior. In particular, it seems that a symmetry break is already present in the "gas-phase" molecule, due to the internal asymmetry in the repeated structural units. In this case, the claim of novelty should be definitely reduced.

In addition, from statistical considerations, the adsorption probability of the molecule on the A and B faces should be the same. Once adsorbed on the Cu surface, due to the large dimensions of the valinomycin, it is unlikely the the molecules can "flip", changing the adsorption configurations. This would lead to an equal mixture of the 2 species on the surfaces. However, the reported measurements indicate an important difference. The authors should comment on the origin of this unbalance.

Minor points:

- 1) labels of the height scales in Fig.1d-2c-3a,d are too small to be clearly read.
- 2) Ref.29 and Ref.40 coincide.

In summary, the paper is worth publishing in Nature Communications provided the above points are clearly addressed.

Reviewer #2 (Remarks to the Author):

The authors have demonstrated that valinomycin can be adsorbed on the surface of Cu crystal in two different configurations: either with its "A-face" up or down. These two configurations have opposite two-dimensional chiralities, but they are not mirror images of one another and have different binding energies. This asymmetry in the binding energies propagates to exclusively produce higher order structures with certain chirality. In particular, there are two levels of organization: first, only right-handed tetramers are formed, and then, the tetramers oligomerize to 2-D supramolecular networks with right-handed chirality. STM images of these supramolecular complexes show excellent agreement with predicted structures from DFT calculations.

I think the results are interesting and worth publishing. However, I have a major concern and a few minor suggestions.

Major Concern:

The problem is that the title, the abstract, the introductory paragraph, and the conclusion of the paper all try to sell this manuscript as a potential mechanism for the origin of biological homochirality. In my opinion, this has nothing to do with biological homochirality and all the references to the subject are better removed. This paper fits in the category of 2-d chiral assembly on metal surfaces. Advertising it as a homochirality paper, in my opinion, attracts the wrong audience who will be disappointed by lack of relevance.

Here is my reasoning for the claim that this paper is not a mechanism for biological homochirality. Valinomycin is a chiral molecule. The experiment uses homochiral valinomycin to start with. The lack of the opposite enantiomer has already broken the symmetry in the initial state. There is no symmetry to be broken in the problem because the two 2-D chiral states are not mirror images of one another and have different binding energies. If the opposite enantiomer of valinomycin were added, for each absorbing configuration, there would be another with the opposite chirality but the same binding energy, and one would expect to see the mirror image of the observed supramolecular network with the same probability (assuming that the order is not destroyed as a result of the interaction of molecules with opposite chirality).

The observed choice of chirality in higher order structures is a result of the fact that the experiment was started with homochiral molecules. Even though it looks like the homochirality is in a different sense, nevertheless, the lack of mirror symmetry in the initial state propagates to this new sense of 2-D chirality.

Minor Comments:

- 1) I think it would be interesting to use the total interaction energies in Table 1 to calculate the probability of observing each tetramer and show some quantitative agreement with the observed counts. It would not be a perfect agreement with the sample size of 166, but it should get close.
- 2) At first glance, Fig. 3e looks like a 2-D hexagonal crystal, but it has a unit cell with the size four times larger. This is because not all the corners of the hexagons are the same tetramers. This is not easy to see in Fig. 3e, but becomes very clear in Fig. 4b. I think this distinction would be interesting to some audience and would be worth mentioning in the paper.
- 3) The paragraph before the summary, at multiple points, claims that this experiment demonstrates that homochirality can emerge from initial non-chiral biomolecules. This is not true; valinomycin is chiral.
- 4) The symbol T_R is used a few lines before it was defined as right-handed chiral tetramer. The definition should be moved up to its first use.
- 5) At a couple of points in the paper, the authors appear to claim that parity violation in weak interaction or coupling with another external chiral field are the only known mechanisms for homochirality. I would like to point out that there is a huge class of spontaneous symmetry breaking mechanisms for homochirality that do not rely on a parity violation. See Section IB of Phys. Rev. E 95, 032407 (2017).
- 6) There were a few confusingly worded sentences in the first couple of pages that are worth cleaning up for clarity.

Reviewer #3 (Remarks to the Author):

The paper presents novel research highlighting a possible mechanism to obtain large homochiral molecular assemblies from a bottom-up approach.

After establishing that there is an excess of right-handed valinomycin molecules on a copper Cu(111) substrate, the authors build right-handed tetramers (TR) of valinomycin out of "left" and "right"-handed valinomycin-molecule monomers on the same substrate. They complement their STM images with density functional theory calculations, showing that, indeed, larger adsorption energies between the tetramers and the Cu(111) lead to more efficient binding and, thus, to a selection rule where only right-handed tetramers are formed out of 166 tetramers.

The authors then take this argument further, showing that TR guide the formation of homochiral supramolecular networks which, again, are chirally-pure. They suggest that the "directional right-attacked" site-binding ensures the selection of chirally-pure supramolecular networks. In essence, they propose a bottom-up mechanism ("hierarchical assembly") that starts with TR's dominance and goes up from there.

In my view, the science is sound and the results solid. It should be possible for another group to reproduce the work, although I am not an experimentalist. I do have, however, a few questions that the authors should address before I would recommend the manuscript for publication.

First, it is not clear to me why the exclusive assembly of TR takes place. One would expect, statistically, that a few TL would form? The results say that no TL form, out of 166 tetramers. Do the authors have an explanation for this absolute dominance?

The statement in the summary paragraph, "We found a novel possible way for the origin of biomolecular homochirality" seems to be unjustified. The authors found a way for biohomochirality to grow from single molecule to large assembly on a very specific substrate and with a molecule with very specific chiral signature. It is not clear to me that the prebiotic mechanism(s) that selected homochirality had anything to do with this one, and I don't see where the authors provide evidence that it does. How general they believe this hierarchical assembly mechanism to be? Would it apply to other chiral molecules and substrates? Can they predict candidates based only on DFT calculations of adsorption energy? Furthermore, they do not offer any arguments justifying valinomycin as a molecule of prebiotic relevance.

As is, the paper seems to be of more relevance to the community interested in developing artificial biohomochiral materials.

Note also that Eq. 3 is hardly legible.

Response letter

Point-by-point response to the referees' comments

Reviewer #1 (Remarks to the Author):

In the submitted manuscript, Chen et al. reported the formation of homochirality upon adsorption of valinomycin molecules on a Cu(111) surface. The authors claimed to observe a difference in population between the two possible molecular adsorption configurations (A and B faces), leading to the formation of a predominant homochiral species, mimicking the origin of chirality in living organisms.

The initial symmetry breaking is then followed by hierarchical growth of a supramolecular assembly of homochiral structures, revealing how the monomer chirality unbalance is amplified in more complex aggregates.

Despite chirality expression on surfaces has been extensively investigated in the last decades by means of surface science techniques, the reported results seem to bring some new inputs in such research field, in particular for their importance in biological systems.

In summary, the paper is worth publishing in Nature Communications provided the above points are clearly addressed.

Author Response: We greatly appreciate the reviewer's positive comment.

1. However, there are a few major discrepancies in the text, that need to be clarified.

Firstly, the formation of chirality through an achiral organic molecule adsorbed on an achiral surface is somehow surprising, probably representing the more interesting results in the whole manuscript. Nevertheless, the authors did not properly comment on the origin of such behavior.

In particular, it seems that a symmetry break is already present in the "gas-phase" molecule, due to the internal asymmetry in the repeated structural units. In this case, the claim of novelty should be definitely reduced.

Author Response: Thanks for the reviewer's suggestions. Chirality is not only originated from the molecular chirality caused by chiral central atom but also from the structural chirality in the topological geometric space such as DNA helical structure, right-handed helix snail and trumpet shells. Different from the concept of "molecular chirality" in organic chemistry; "structural chirality" emphasizes on the helical structures in topological geometric space rather than an individual chiral central atom. In our case, the surface-mediated 2D chirality is a "structural chirality".

In gas phase, valinomycin has not any defined “structural chirality” since the molecules may flip or distort randomly. However, when they are adsorbed on the Cu(111) surface, the conformation of valinomycin will be immobilized by the surface. When the “A-face” or “B face” of the bowl-shaped valinomycin molecules landing on the Cu(111) surface, the topological geometric arrangement of three tetrapeptide units with asymmetrical structure appear as counterclockwise or clockwise propellers, expressed as a pair of enantiomers with inverse surface-mediated “structural chirality”. The probabilities to land on the surface via “A face” or “B face” are the same for gaseous valinomycin, and no symmetry break of “structural chirality” happens in the “gas-phase” molecules. Nevertheless, the symmetry break of “structure chirality” appears with the aid of the surface adsorption. The difference in adsorption-desorption thermodynamics results in the quantities unbalance of the two adsorbed configurations with different “structural chirality” (M_L and M_R). Surface adsorption plays a key role in the origin of “structural chirality” and symmetry break of the chirality. To avoid misunderstanding, we have modified “a biomolecule without a defined chirality” to “a biomolecule without a defined “structural chirality”” in the revised manuscript. Our understanding about the origin of “structural chirality” and symmetry break has further been stated in the revised manuscript.

2. In addition, from statistical considerations, the adsorption probability of the molecule on the A and B faces should be the same. Once adsorbed on the Cu surface, due to the large dimensions of the valinomycin, it is unlikely the molecules can "flip", changing the adsorption configurations. This would lead to an equal mixture of the 2 species on the surfaces. However, the reported measurements indicate an important difference. The authors should comment on the origin of this unbalance.

Author Response: Though the probabilities to fall on the surface from their “A face” or “B face” are the same, the adsorption and desorption balance will affect the ultimate number of the two adsorption configurations on the surface. The adsorption energies calculated by density function theory (DFT) calculation indicate that M_R (-39.646 kcal/mol) is more stable than M_L (-33.895 kcal/mol). Desorption of M_L may occur more easily than M_R , which results in the higher amount of M_R on the surface than M_L . The difference in adsorption-desorption thermodynamics results in the quantities unbalance of the two adsorption configurations and the symmetry break of the chirality.

Minor points:

1) Labels of the height scales in Fig.1d-2c-3a, d are too small to be clearly read.

Author Response: Thanks for the reviewer’s suggestion and we have enlarged them in the revised manuscript.

2) Ref.29 and Ref.40 coincide.

Author Response: Thanks for the reviewer’s reminder and we have modified them in the revised manuscript.

Reviewer #2 (Remarks to the Author):

The authors have demonstrated that valinomycin can be adsorbed on the surface of Cu crystal in two different configurations: either with its "A-face" up or down. These two configurations have opposite two-dimensional chiralities, but they are not mirror images of one another and have different binding energies. This asymmetry in the binding energies propagates to exclusively produce higher order structures with certain chirality. In particular, there are two levels of organization: first, only right-handed tetramers are formed, and then, the tetramers oligomerize to 2-D supramolecular networks with right-handed chirality. STM images of these supramolecular complexes show excellent agreement with predicted structures from DFT calculations.

I think the results are interesting and worth publishing. However, I have a major concern and a few minor suggestions.

Author Response: We greatly appreciate the reviewer's positive comment.

Major Concern:

The problem is that the title, the abstract, the introductory paragraph, and the conclusion of the paper all try to sell this manuscript as a potential mechanism for the origin of biological homochirality. In my opinion, this has nothing to do with biological homochirality and all the references to the subject are better removed. This paper fits in the category of 2-d chiral assembly on metal surfaces. Advertising it as a homochirality paper, in my opinion, attracts the wrong audience who will be disappointed by lack of relevance.

Here is my reasoning for the claim that this paper is not a mechanism for biological homochirality. Valinomycin is a chiral molecule. The experiment uses homochiral valinomycin to start with. The lack of the opposite enantiomer has already broken the symmetry in the initial state. There is no symmetry to be broken in the problem because the two 2-D chiral states are not mirror images of one another and have different binding energies. If the opposite enantiomer of valinomycin were added, for each absorbing configuration, there would be another with the opposite chirality but the same binding energy, and one would expect to see the mirror image of the observed supramolecular network with the same probability (assuming that the order is not destroyed as a result of the interaction of molecules with opposite chirality). The observed choice of chirality in higher order structures is a result of the fact that the experiment was started with homochiral molecules. Even though it looks like the homochirality is in a different sense, nevertheless, the lack of mirror symmetry in the initial state propagates to this new sense of 2-D chirality.

Author Response: Thanks for the reviewer's suggestions. Biological homochirality not only exist in the origin of life but also in the advanced biological tissues and organisms. For example, the geometric structures of many living organisms always exhibit homochirality, such as right-handed helix snail shells. The concept of

“biological homochirality” is not merely limited to the origin of life but also applicable to natural biological tissues/organs and artificial structures/materials. We agree with the review that more general description could possibly bring broader readers lack of relevance, thus we have changed the title to make the manuscript more specific. In the revised manuscript, we focused the concept of “biological homochirality” on the homochiral biological structures and materials. The title, the abstract, the introductory paragraph, the conclusion and the references have been carefully readjusted in the revised manuscript.

Chirality is not only originated from “molecular chirality” caused by chiral central atom but also from “structural chirality” in the topological geometric space such as DNA helical structure, right-handed helix snail and trumpet shells. Different from the concept of “molecular chirality” in organic chemistry; “structural chirality” emphasizes on the helical structures in topological geometric space rather than an individual chiral central atom. In our manuscript, the surface-mediated chirality is a “structural chirality”. The hierarchical assembly starts by using M_L and M_R as the initial building blocks. It is worth mentioning that the homochirality in the hierarchical assembly is originated from the chirality unbalance of two adsorption configuration monomers (M_L and M_R), rather than starts from the gaseous valinomycin. M_L and M_R are enantiomers whose chiral symmetric mirror is parallel to the Cu(111) surface. In other words, the origin of surface-mediated homochirality in the hierarchical assembly doesn't start from homochiral building blocks but a pair of enantiomers.

Though the probabilities to fall on the surface from their “A face” or “B face” are the same, the adsorption and desorption balance will affect the ultimate number of the two adsorption configurations (M_L and M_R) on the surface. The adsorption energies calculated by density function theory (DFT) calculation indicate that M_R (-39.646 kcal/mol) is more stable than M_L (-33.895 kcal/mol). Desorption of M_L may occur more easily than M_R , which results in the higher amount of M_R on the surface than M_L . The difference in adsorption-desorption thermodynamics results in the quantities unbalance of M_L and M_R . That is the initial symmetry break of the chirality.

Figure 4c demonstrates the complete evolution process of the surface-mediated homochirality in the hierarchical assembly, starting from the enantiomers (M_L and M_R). Asymmetrical adsorption of the nonplanar valinomycin molecules on the Cu(111) surface leads to unequal population of M_L and M_R , which is the initial building blocks for the hierarchical assembly. The initial symmetry break is amplified by the directional site-specific recognition between these two heterochiral monomers at the first level assembly, leading to the formation of homochiral tetramers (T_R). The homochirality is further transferred when these homochiral tetramers assemble into the homochiral supramolecular networks (S_R) at the second level assembly. Our manuscript demonstrates a complete homochirality evolution process, including symmetry break, chirality amplification and chirality transmission. Therefore, we think that our work not merely demonstrated a 2D chiral assembly but also revealed

the origin and evolution mechanisms of surface-mediated homochirality in the hierarchical assembly of biomolecules.

Minor Comments:

1) *I think it would be interesting to use the total interaction energies in Table 1 to calculate the probability of observing each tetramer and show some quantitative agreement with the observed counts. It would not be a perfect agreement with the sample size of 166, but it should get close.*

Author Response: The probability of tetramer is the statistics results of experimental observations. The calculated total interaction energies indicate that T_R is more stable than T_L , which can qualitatively agree with the experimental results. We have checked again and found that only T_R was observed in our experiments. It may be possible that a few of T_L exists beyond our observations.

2) *At first glance, Fig. 3e looks like a 2-D hexagonal crystal, but it has a unit cell with the size four times larger. This is because not all the corners of the hexagons are the same tetramers. This is not easy to see in Fig. 3e, but becomes very clear in Fig. 4b. I think this distinction would be interesting to some audience and would be worth mentioning in the paper.*

Author Response: Thanks for the reviewer's suggestion. To make it clearer, a unit cell of supramolecular networks was superimposed by simplified monomer symbols in Fig. 3e. We also have also added some text description in the revised manuscript to describe this interesting distinction.

3) *The paragraph before the summary, at multiple points, claims that this experiment demonstrates that homochirality can emerge from initial non-chiral biomolecules. This is not true; valinomycin is chiral.*

Author Response: Thanks for the reviewer's reminder. Chirality includes "molecular chirality" and "structural chirality". In our case, the surface-mediated 2D chirality is a "structural chirality". We fully agree with the reviewer that valinomycin is chiral because of the existence of chiral center atoms. While in gas phase, valinomycin has not any defined "structural chirality", since the molecules may flip or distort randomly. For better clarify, we have changed the description of the gaseous valinomycin as "no defined structural chirality" in the revised manuscript.

4) *The symbol T_R is used a few lines before it was defined as right-handed chiral tetramer. The definition should be moved up to its first use.*

Author Response: Thanks for the reviewer's careful reading. We have modified it in the revised manuscript.

5) *At a couple of points in the paper, the authors appear to claim that parity violation in weak interaction or coupling with another external chiral field are the only known mechanisms for homochirality. I would like to point out that there is a huge class of spontaneous symmetry breaking mechanisms for homochirality that do not rely on a*

parity violation. See Section IB of Phys. Rev. E 95, 032407 (2017).

Author Response: We are grateful for the kind suggestions. We have removed the concept of “biological homochirality” away from the origin of life according to reviewers’ comments, but only focused on the biological structures/materials in the revised manuscript.

6) *There were a few confusingly worded sentences in the first couple of pages that are worth cleaning up for clarity.*

Author Response: We are grateful for the kind reminder. We have carefully checked and modified them again.

Reviewer #3 (Remarks to the Author):

The paper presents novel research highlighting a possible mechanism to obtain large homochiral molecular assemblies from a bottom-up approach.

After establishing that there is an excess of right-handed valinomycin molecules on a Cu(111) substrate, the authors build right-handed tetramers (T_R) of valinomycin out of “left” and “right”-handed valinomycin-molecule monomers on the same substrate. They complement their STM images with density functional theory calculations, showing that, indeed, larger adsorption energies between the tetramers and the Cu(111) lead to more efficient binding and, thus, to a selection rule where only right-handed tetramers are formed out of 166 tetramers.

The authors then take this argument further, showing that T_R guide the formation of homochiral supramolecular networks which, again, are chirally-pure. They suggest that the “directional right-attacked” site-binding ensures the selection of chirally-pure supramolecular networks. In essence, they propose a bottom-up mechanism (“hierarchical assembly”) that starts with T_R ’s dominance and goes up from there.

In my view, the science is sound and the results solid. It should be possible for another group to reproduce the work, although I am not an experimentalist. I do have, however, a few questions that the authors should address before I would recommend the manuscript for publication.

As is, the paper seems to be of more relevance to the community interested in developing artificial biohomochiral materials.

Author Response: We greatly appreciate the reviewer’s positive comment.

1. First, it is not clear to me why the exclusive assembly of T_R takes place. One would expect, statistically, that a few T_L would form? The results say that no T_L form, out of 166 tetramers. Do the authors have an explanation for this absolute dominance?

Author Response: Thanks for the reviewer's kind suggestion. We have checked again and found that only T_R was observed in our experiments. It may be possible that a few of T_L exists beyond our observations. Valinomycin tetramers whether T_L or T_R , are the co-assembled products of two kinds of adsorbed conformation monomers (M_L and M_R). Different from the undisturbed self-assembly, the formation of T_L and T_R are competitive. Our calculated results in Table 1 indicate that the interaction energy among monomer subunits in the formation of T_R is far higher than that of T_L . The directional site-binding in the "right-attacked" chiral recognition becomes the predominant chiral recognition mode, and T_R becomes the dominant assembly products. We believe that the spatial conformational complementary and the strong intermolecular interactions play an important role in the formation of homochiral tetramers.

2. *The statement in the summary paragraph, "We found a novel possible way for the origin of biomolecular homochirality" seems to be unjustified. The authors found a way for biohomochirality to grow from single molecule to large assembly on a very specific substrate and with a molecule with very specific chiral signature. It is not clear to me that the prebiotic mechanism(s) that selected homochirality had anything to do with this one, and I don't see where the authors provide evidence that it does. How general they believe this hierarchical assembly mechanism to be? Would it apply to other chiral molecules and substrates? Can they predict candidates based only on DFT calculations of adsorption energy? Furthermore, they do not offer any arguments justifying valinomycin as a molecule of prebiotic relevance.*

Author Response: Biological homochirality not only exist in the origin of life but also in the advanced biological tissues/organs and artificial structures/materials. For example, the geometric structures of many living organisms always exhibit homochirality, such as right-handed helix snail shells. In the revised manuscript, we focused on homochiral biological structures and materials. The expression of "We found a novel possible way for the origin of biomolecular homochirality" was updated to "We found a novel possible trigger way for the formation of advanced homochiral biological structures and materials" in the revised manuscript. In our case, valinomycin was selected as a scientific model to understand the surface-mediated homochirality. We agreed that the case of valinomycin can provide some helpful inspiration for the fabrication of other 2D homochiral complex materials.

Our work gave out the single-molecule evidence that hierarchical assembly is a significant strategy for the self-organization of highly ordered biological structures guided by the homochirality selection rule. Since the intermolecular interactions play a pivotal role in the hierarchical assembly of valinomycin, other suitable substrates should be also applicable thus the hierarchical assembly strategy can be propagated into the field of fabrication of versatile two-dimensional (2D) artificial homochiral molecular structures and materials. Density function theory (DFT) calculations are helpful to predict the candidate molecules and their suitable substrates. The candidate molecules may possess the following structural features: (1) the molecules have

multiple binding sites and different symmetries for hierarchical assembly (reference: Chem Soc Rev 2009, 38: 2576); (2) the molecules have two different adsorption faces with diverse chemical composition and structure, so that it is possible to obtain two adsorption configurations when they adsorbed on the surface; (3) the molecules have a multi-fold symmetric axis perpendicular to their adsorption faces and the repeated units has asymmetrical inner structures, so that they can produce “structural chirality” when they adsorbed on the surface.

3. Note also that Eq. 3 is hardly legible.

Author Response: We have revised the description for DFT-D3 method. In the DFT-D3 correction method of Grimme et al. (reference: J Chem Phys 2010, 132: 154104), the following vdW-energy expression is used:

$$E_{\text{disp}} = -\frac{1}{2} \sum_{i=1}^{Nat} \sum_{j=1}^{Nat} \sum_L \left(f_{d,6}(r_{ij,L}) \frac{C_{6ij}}{r_{ij,L}^6} + f_{d,8}(r_{ij,L}) \frac{C_{8ij}}{r_{ij,L}^8} \right) \quad (2)$$

where $r_{ij,L}$ is the internuclear distance. The dispersion coefficients C_{6ij} and C_{8ij} are geometry-dependent as they are adjusted on the basis of local geometry (coordination number) around atoms i and j . The Becke-Jonson (BJ) damping can be used in the DFT-D3 method:

$$f_{d,n}(r_{ij}) = \frac{s_n r_{ij}^n}{r_{ij}^n + (a_1 R_{0ij} + a_2)^n} \quad (3)$$

Where $R_{0ij} = \sqrt{C_{8ij} / C_{6ij}}$, the parameters a_1 , a_2 , s_6 , and s_8 are adjustable depending on the choice of exchange-correlation functional.

In addition, the modifications in the figures were summarized below.

1. We updated the label of the height scales of STM images (Figure 1d, 2a, 2c, 3a, and 3d), so that the words become big enough to read.
2. We overlapped simplified monomer symbols on the model of the supramolecular networks in Figure 3e; as a result, the type and quantum of molecules in a unit cell can be quickly found.
3. To better understand the homochirality evolution processes in the hierarchical assembly, some text descriptions were added in Figure 4.

Reviewers' comments:

Reviewer #1 (Remarks to the Author):

After revision, the manuscript quality is improved in many points. However, in my opinion, a major issue has not been properly addressed yet. Indeed, in general, a molecule without "any defined structural chirality" adsorbed on an achiral surface, such as Cu(111), has no reason to exhibit homochirality. For the case of valinomycin, for example, it is not clear how the arrangement of the tetrapeptide units is only counterclockwise for the "A-face". For symmetry reasons, there should also be exactly the same number of molecules in the clockwise "A-face". The same reasoning holds for the "B-face". If not, what is breaking the symmetry?

The authors should provide a clear and convincing explanation, other than DFT energetics. This point is the key to understand the whole paper and I believe that currently it lacks of a reasonable explanation, resulting confusing for the majority of readers.

Reviewer #2 (Remarks to the Author):

All points raised in my previous report have been addressed and the paper can be published in its present form.

Reviewer #3 (Remarks to the Author):

I have read the revised manuscript and the responses to all three reviews. The authors have complied with the suggestions from all of us in what I consider a satisfactory fashion. It is my opinion that this manuscript is now acceptable for publication in Nature Communications.

Response letter

Point-by-point response to the referees' comments

Reviewer #1 (Remarks to the Author):

After revision, the manuscript quality is improved in many points. However, in my opinion, a major issue has not been properly addressed yet. Indeed, in general, a molecule without "any defined structural chirality" adsorbed on an achiral surface, such as Cu(111), has no reason to exhibit homochirality. For the case of valinomycin, for example, it is not clear how the arrangement of the tetrapeptide units is only counterclockwise for the "A-face". For symmetry reasons, there should also be exactly the same number of molecules in the clockwise "A-face". The same reasoning holds for the "B-face". If not, what is breaking the symmetry?

The authors should provide a clear and convincing explanation, other than DFT energetics. This point is the key to understand the whole paper and I believe that currently it lacks of a reasonable explanation, resulting confusing for the majority of readers.

Author Response: We greatly appreciate the reviewer's positive comment. We agree that it has been rarely reported that a molecule without "molecular chirality" or "structural chirality" in three dimensional space exhibits two dimensional (2D) "structural chirality" when they adsorbed on achiral surfaces. When the 2D "structural chirality" originates from the adsorption-induced structural asymmetry in spatial configurations, an achiral surface will be also feasible (reference: Angew. Chem. Int. Ed. 2005, 44: 5334). Moreover, homochiral may occur if one adsorption configuration has an absolute advantage in energy over other adsorption configurations. Besides 2D "structural chirality", adsorption-induced electronic structural chirality has been also reported (reference: Phys. Rev. Lett. 2010, 105: 115702).

For the case of valinomycin, the adsorption-induced 2D structural chirality is determined by its intrinsic asymmetrical chiral tetrapeptide units and it doesn't depend on the chirality of the surface. DFT calculations reveal that three intramolecular N-H...O hydrogen bonds form between the carbonyl oxygen of *L*-Lac residue and the amine group of *D*-Val residue (Fig. S1). The energy of a hydrogen bond is 9.34 kcal·mol⁻¹ and the H...O distance is 1.81 Å. Hydrogen bonds are strong enough to stabilize the framework of the cyclic main chain in the form of a shallow bowl. The side chains lie on the exterior of the bowl with a certain orientation, which is determined by the chirality of the amino acid residues (*L*-Val, *D*-Hyv, *D*-Val and *L*-Lac). Such spatial restriction for valinomycin leads to an unambiguous counterclockwise configuration when valinomycin adsorbed on the surface via the "A face". Clockwise configuration is hard to appear for the case of "A face" since the

change of conformation through rotating chemical bonds will be restricted by the intramolecular hydrogen bond interactions and the inflexibility of cyclic main chain. Similarly, the arrangement of three asymmetric tetrapeptide structural units is clockwise, if valinomycin adsorbed on the Cu(111) surface via the “B face”.

Obviously, a gaseous valinomycin has not defined “structural chirality” when the chiral reference (the Cu(111) surface) is absent, since it flips randomly in three dimensional (3D) space. Only when valinomycin adsorbed on the surface, it will be immobilized with an unambiguous configuration under the surface confinement. In a word, surface adsorption plays a key role in the formation of 2D “structural chirality” and the “structural chirality” is determined by its intrinsic asymmetrical chiral tetrapeptide units.

Reviewer #2 (Remarks to the Author):

All points raised in my previous report have been addressed and the paper can be published in its present form.

Author Response: We greatly appreciate the reviewer’s positive comment.

Reviewer #3 (Remarks to the Author):

I have read the revised manuscript and the responses to all three reviews. The authors have complied with the suggestions from all of us in what I consider a satisfactory fashion. It is my opinion that this manuscript is now acceptable for publication in Nature Communications.

Author Response: We greatly appreciate the reviewer’s positive comment.

REVIEWERS' COMMENTS:

Reviewer #1 (Remarks to the Author):

The authors addressed the remaining issue. The paper can be published as is.